# Preparation and Hydration Mechanisms of Low Carbon Ferrochrome Slag-Granulated Blast Furnace Slag Composite Cementitious Materials

**DOI:** 10.3390/ma16062385

**Published:** 2023-03-16

**Authors:** Chao Ren, Keqing Li, Yonghua Wang, Yanfang Li, Jiannan Tong, Jingyao Cai

**Affiliations:** 1School of Civil Engineering, Tangshan University, Tangshan 063000, China; 2Hebei Key Laboratory of Construction Engineering and Tailings Comprehensive Utilization, Tangshan 063000, China; 3School of Civil and Resources Engineering, University of Science and Technology Beijing, Beijing 100083, China

**Keywords:** low carbon ferrochrome slag, solid waste, composite cementitious materials, compressive strength, hydration characteristics

## Abstract

Low carbon ferrochrome slag (LCFS) is the metallurgical waste slag from the carbon ferrochrome alloy smelting process. Compared with high carbon ferrochrome slag, LCFS has great potential as cementitious material; the chemical compositions of the two types of slag are quite different. In this research, composite cementitious materials are prepared which use low carbon ferrochrome slag and granulated blast furnace slag (GBFS) as the main raw material. Steel slag mud (SSM) and flue gas desulfurization gypsum (FGDG) are used as the activator. In order to find the variety rule of compressive strength on the composite cementitious materials, a three-factor three-level Box-Behnken design is used to discuss the following independent variables: LCFS content, GBFS content, and water-binder ratio. Moreover, the hydration characteristics of the LCFS-GBFS composite cementitious materials is studied in this paper in terms of hydration product, micromorphology, and hydration degree, based on multi-technical microstructural characterizations. The results show that the compressive strength of the LCFS-GBFS composite cementitious materials is significantly affected by single factors and the interaction of two factors. The mechanical property of the mortar samples at 3, 7, and 28 days are 26.6, 35.3, and 42.7 MPa, respectively, when the LCFS-GBFS-SSM-FGDG ratio is 3:5:1:1 and the water-binder ratio is 0.3. The hydration products of LCFS-GBFS composite cementitious materials are mainly amorphous gels (C-S-H gel), ettringite, and Ca(OH)_2_. With the increase of LCFS content, more hydration products are generated, and the microstructure of the cementitious system becomes more compact, which contributes to the compressive strength. The results of this research can provide a preliminary theoretical foundation for the development of LCFS-GBFS composite cementitious materials and promote the feasibility of its application in the construction industry. Deep hydration mechanism analysis and engineering applications should be studied in the future.

## 1. Introduction

Ferrochrome slag is a gray-black metallurgical slag which is produced during smelting carbon ferrochrome alloy with different carbon content. The metallurgical waste slag is formed after high temperature reduction of chromite by pyroelectric furnace at 1700 °C which uses carbon as a reducing agent [1]. According to the carbon content, this can be divided into low carbon and high carbon ferrochrome slag. The high carbon ferrochrome slag is generally agglomerate, and the low carbon ferrochrome slag usually consists of small particles. In 2022, 14.62 million tons of carbon ferrochrome slags were produced internationally. Such a vast amount of carbon ferrochrome slags occupies the land and pollutes the surrounding environment, such as groundwater and vegetation.

In order to achieve the purpose of reducing chromium and lowering the SiO_2_ of the low carbon ferrochrome slag, a large amount of high purity CaO was added during the smelting process. Figure 1 shows the production process of low carbon ferrochrome slag. The chemical composition of the slag system is mainly CaO and SiO_2_, followed by Al_2_O_3_, MgO, and Cr_2_O_3_, and a small amount of Fe_2_O_3_. The main mineral composition in the slag is diacalcium silicate (2CaO·SiO_2_), followed by a small quantity of magnesium rose olivine (3CaO·MgO·SiO_2_) and calcium-aluminite (2CaO·Al_2_O_3_·SiO_2_). Therefore, the low carbon ferrochrome slag has high potential activity and available value [2,3,4]. The chemical composition of the high carbon ferrochrome slag is mainly SiO_2_, Al_2_O_3_, and MgO. The content of CaO is relatively low; however, the content of Al_2_O_3_ and MgO is high. The mineral phases of the slag are mainly magnesium olivine (MgSiO4), magnesium-chromium spinel [Mg(Al_1.5_Cr_0.5_)O_4_], and magnesium-aluminum spinel (MgAl_2_O_4_). Therefore, the high carbon ferrochrome slag has low activity, and it can be used as concrete aggregate [5,6,7,8,9]. It has been shown that, by adjusting the firing temperature and the composition of raw materials, the spinel phase in ceramics can reduce the leaching of Cr very well [10]. At present, scholars all over the world have undertaken preliminary research on the utilization of ferrochrome slag. Using FCA instead of blast furnace slag, Omur [11] showed that the replacement amount is 30%, and studied the possibility of its being the most alkali excitation material. Khabibulin [12] prepared the concrete by using Chrome slag, fly ash and cement as raw materials. Acharya [13] reviewed the preparation of concrete from chromite slag, and reported its work characteristics, economic advantages, and environmental benefits. Isil [14] evaluates the feasibility of ferrochrome slag as cementing material, and studied the effect of it as concrete admixture on mechanical properties. Nath [15] investigated the preparation of cement-based materials from chromite slag and fly ash. However, there is little research on the preparation of new composite cementitious materials with no cement, which uses the low carbon ferrochrome slag as raw material.

The granulated blast furnace slag is the industrial waste which is discharged in the process of iron smelting. As a typical material which has pozzolanic activity, granulated blast furnace slag is formed after water quenching and extremely cold treatment, it has vitreous structure and potential activity which is widely used in the cement and concrete industry. Researchers have carried out many experiments and suggested engineering applications of the granulated blast furnace slag [16,17,18,19,20]. Dai [21] analyzed the relationship between the porosity of cement slurry and the specific surface area of slag, slag content, and curing age. Li [22] used steel slag to replace part of the slag, and sodium sulfate was used to stimulate the activity of the slag—slag gelling system. Park [23] investigated the effects of SCMs, expander, and crystallizer on the hydration products of cementified materials. These views clarify the hydration reaction characteristics of granulated blast furnace slag.

Steel slag mud is the secondary waste slag which is generated from steel slag by wet magnetic separation. The high active component of the steel slag mud has hydration reaction, which leads to difficulty of utilization. At present, there are few studies on steel slag mud. However, most scholars have made some achievements in the study of steel slag [24,25,26,27,28].

The main component of flue gas desulfurization gypsum is calcium sulfate dihydrate which is the same as natural gypsum. A large number of experimental results [29,30,31,32] show that flue gas desulfurization gypsum can be used as an important activator in cement-based materials.

At present, the research of new cementitious materials as a substitute for cement has become a hot topic for researchers all over the world [33,34,35,36]. The LCFS are commonly used as a cement admixture and be added in very small amounts. However, there are few studies on cement-free cementitious materials by using LCFS as raw material. In this paper, the LCFS-GBFS composite cementitious materials were prepared by using LCFS, GBFS, SSM, FGDG as raw materials. Moreover, the influence on the variation of compressive strength of the LCFS-GBFS composite cementitious materials is studied according to different factors. In order to analyze the hydration products and hydration mechanism of the LCFS-GBFS composite cementitious materials, and provide the theoretical basis for improving the utilization rate of LCFS, various analytical and testing methods are used. It is worth noting that the hydration mechanism and mechanical properties of LCFS-GBFS composite cementitious materials were only preliminarily studied, and there are many deficiencies. Therefore, more extensive and in-depth studies on the hydration mechanism of LCFS-GBFS composite cementitious materials will be carried out in the future, which includes the experiment of single mineral in LCFS and the synergistic reaction mechanism of multiple raw materials. Moreover, a large number of construction applications should be considered so that the durability of the LCFS-GBFS composite cementitious materials can be tested.

## 2. Materials and Methods

### 2.1. Raw Materials

In this study, the low carbon ferrochrome slag is produced from a metallurgical factory in Inner Mongolia, China. The low carbon ferrochrome slag was grinded into micronized powder through laboratory ball miss for 100 min. The chemical composition of LCFS is presented in Table 1. The X-ray diffraction (XRD) result of LCFS is shown in Figure 2a. It can be seen that the main phases in LCFS are C_2_S, MgO, and Ca_2_Al_2_O_4_. As shown in Figure 3a, the gap of the sample is filled with many flocculent substances. The EDS result indicates that the presence of C_2_S may be due to the element of the flocculent substances O, Si, and Ca.

The granulated blast furnace slag and steel slag mud used are obtained from Shougang iron and steel Co., Ltd., Beijing, China. The granulated blast furnace slag and steel slag mud were grinded into micronized powder through laboratory ball miss for 80 min. The chemical compositions are listed in Table 1. The mineral compositions of GBFS and SSM are analyzed using XRD as shown in Figure 2b,c. The main mineral phases of SSM, including C_2_S, C_3_S, C_2_F, and Ro phases, are detected, and the absence of distinct sharp peaks in Figure 2c indicate that GBFS has an amorphous characteristic structure.

The flue gas desulfurization gypsum is provided by Tangshan power company. The flue gas desulfurization gypsum was grinded into micronized powder through laboratory ball miss for 30 min. Table 1 lists the chemical composition of FGDF, and the XRD result of FGDG is shown in Figure 2d. It can be seen the main phase is gypsum.

The specific surface areas of LCFS, GBFS, SSM, and FGDG are 915 m^2^/kg, 667 m^2^/kg, 530 m^2^/kg, and 817 m^2^/kg. The particle size distribution of the raw materials and the percentage of particle size are shown in Figure 4. It can be seen the median particle size (D_50_) of LCFS, GBFS, SSM, and FGDG are 6.035 μm, 10.10 μm, 47.12 μm, and 6.515 μm. The LCFS has a smaller D_50_, and the distribution proportion of micro particle size is larger; therefore, this is beneficial by bringing into play the hydration and filling effect of micro particles.

### 2.2. Mix Proportions

The mix proportions of LCFS-GBFS cementitious materials are shown in Table 2. The water-binder ratio of LCFS-GBFS cementitious materials pastes prepared is 0.35. For mortar, the water-binder ratio is 0.3, and the binder-sand ratio is 1:3. The mix proportion of mortars prepared includes 450 g of cementitious and 1350 g of sand. For mixing mortar, the LCFS, GBFS, SSM, FGDG, and sand are first poured into the bowl under a low-speed mixing for 30 s, then water is poured under a low-speed mixing for 90 s. Finally, the mixture is mixed under a high-speed for 120 s. The process of mixing the paste is the same as the mixing of the mortar, except no sand is used. The pastes are cast into 30 mm × 30 mm × 50 mm molds for microstructural analysis when the measuring age at 3, 7, and 28 days. The mortar is cast into 40 mm × 40 mm × 60 mm mold for the mechanical tests with the same measuring age as pastes.

### 2.3. Experimental Methods

#### 2.3.1. Compressive Strength

The mortar and paste were poured into the molds of size as 40 mm × 40 mm × 60 mm and 30 mm × 30 mm × 50 mm respectively and test strength, and the samples were cured at 20 ± 1 °C, and >95% humidity for 3, 7, and 28 days respectively. The strengths were measured according to the China standard GB/T 17671-1999, and the values of the three samples were reported as the compressive strength.

#### 2.3.2. Design of Experiment Using Response Surface Methodology

Box-Behnken design of response surface is a common experimental design method which is widely used. In this paper, a three-level three-factor Box-Behnken design with Design-Expert software (v8.0.6.1) is employed. The contents of LCFS, GBFS, and water-binder ratio were taken as independent factors, and expressed by X_1_, X_2_, and X_3_ respectively. The compressive strength of mortar samples at 3, 7, and 28 days was used as response value, and expressed by Y_1_, Y_2_, and Y_3_ respectively. The code and level of response surface design factors are shown in Table 3.

#### 2.3.3. X-ray Diffraction

The polycrystalline X-ray diffractometer (D/MAX-RC, Rigaku, Tykyo, Japan) equipped with Cu-Kα radiation at 40 kV and 40 mA was used to analyze mineral composition of hydration products at 28 days with an angular accuracy of ±0.01°, a scanning speed of 5°/min, and a scanning range of 10–80°.

#### 2.3.4. SEM Analysis

SEM technique (scanning eceltron microscope, Hitashi SU8820, Tokyo City, Japan) was used to investigate the microstructure of paste samples. The micro-morphology of samples was imaged under the condition of 15 kV accelerating voltage. EDS testing was also conducted to analyze the chemical composition of hydration products.

#### 2.3.5. Hydration Heat

The hydration heat process of the LCFS-GBFS composite cementitious materials was examined by an isothermal calorimeter with a sensitivity of 0.4 μJ and baseline stability of ±0.08 μW/h. The measured data were recorded automatically by the instrument. All experiments were conducted at 25 °C, and the sample was prepared with a water-binder at 0.4 because of the good flowability of the blended paste.

#### 2.3.6. Fourier Transform Infrared Spectroscopy

The chemical structure of minerals for the hydration products in the paste samples was tested using Fourier transform infrared spectroscopy (FTIR). A NEXUS670 FTIR infrared spectrometer was operated for FTIR analysis, with a 400 to 4000 cm^−1^ wave number and a resolution of 3 cm^−1^.

#### 2.3.7. Solid State Nuclear Magnetic Resonance

^29^Si DD MAS-NMR spectra (119.26 MHz) and ^27^Al MAS-NMR (156.41 MHz) spectra were collected on a solid-state NMR spectrometer to study the polymerization of the hydration products. The spinning rate was 12 kHz, and the repetition delays were 20 and 5 s, respectively.

## 3. Results and Discussions

### 3.1. Compressive Strength of Mortar

The compressive strength of mortar samples at the ages of 3, 7, and 28 days are shown in Figure 5. The results show that the compressive strength of mortar samples at various ages were improved by the LCFS content adding, especially at the early age. At 3 days, the compressive strength of mortar samples increased by 39.4%, 74.7%, 128.5%, respectively, which contained 10%, 20%, 30% LCFS. It indicates that the Ca(OH)_2_ was generated quickly, but the liquid phase still contains many Ca^2+^ and OH^−^, which can promote the decomposition of GBFS, and resulted in the rapid formation of hydration products such as ettringite and C-S-H gels. Moreover, C-S-H gels were also mass generated by C_3_S in LCFS, which filled the pores and enhanced the early compressive strength of mortar samples.

As the hydration age increased, the compressive strength of each group was significantly enhanced. During the curing from 3 days to 28 days, the compressive strength of mortar samples increased by 4.4 MPa, 13.3 MPa, 20.8 MPa respectively, which contained 10%, 20%, 30% LCFS respectively, as presented in Figure 5. It can be seen that the inclusion of LCFS has a positive effect on the strength development of mortar samples. This is because the C_3_S and C_2_S in LCFS can hydrate to form C-S-H gel and Ca(OH)_2_, which is similar to the theory that cement provides compressive strength. Moreover, the liquid phase still contains many Ca^2+^ and OH^−^. The solution was left in a high alkalinity state. These factors contribute to the depolymerization of silico-oxygen tetrahedrons and al-oxygen tetrahedrons in GBFS, and generate a large amount of active (H_2_SiO_4_)^2−^ and (H_2_AlO_3_)^−^. These react with the Ca^2+^ and SO_4_^2−^ in the solution to produce C-S-H gel and ettringite. Therefore, the more LCFS was added, the more significant the hydration of these minerals is, and a large amount of hydration products were generated. This is the theoretical basis of our proposal that an LCFS increase has a positive effect on the compressive strength of mortar samples.

### 3.2. Response Surface Experiment Results and Analysis

Based on the Design-Expert software, the test results of the response surface method were nonlinearly fitted [37]. The compressive strength quadratic polynomial regression equation was established for the model at 28 days. The relationship between the predicted response value (Y) and independent factors can be expressed. It was represented in the following equations:Y = −0.71 + 43.2X_3_ − 0.08775X_1_ + 0.48675X_2_ − 2.775X_1_X_3_ − 0.025X_2_X_3_ + 0.008X_1_X_2_ − 34X_3_^2^ + 0.04285X_1_^2^ − 0.00415X_2_^2^(1)
where X1, X2, and X3 represent the content of LCFS, GBFS, and water-binder ratio.

Figure 6a shows the response surface diagram of the 28-day compressive strength to examine the interactions between the LCFS and GBFS content under the water-binder ratio is 0.3. Moreover, the corresponding contour map of the 28-day compressive strength response surface is shown in Figure 6b. When the water-binder ratio is 0.3, with the increase of the LCFS content, the compressive strength of the mortar samples increased. Meanwhile, the compressive strength increased rapidly and the response surface became steeper when the LCFS content exceeded 20%. When the GBFS content was 50% and the LCFS content increased from 10 to 20%, the compressive strength was an increase of 10.4 MPa, as the LCFS content increased from 20 to 30%, the compressive strength was an increase of 20.9 MPa. As is shown in Figure 6b, the spacing of contour lines were from wide to narrow, which also reflected this trend. When the LCFS content was low, with the increase of GBFS content, the compressive strength of the mortar samples increased, but strength increased slowly. These results were caused by the higher-activity of LCFS hydrated to from C-S-H gels and Ca(OH)_2_. Meanwhile, the GBFS was dissociated, then reacted with FGDG and Ca(OH)_2_ to from C-(A)-S-H gels, ettringite, and other silicaluminates. A large amount of hydration products interlaced and bonded the loose skeleton into a whole unit, which improved the compressive strength of the mortar samples.

Figure 7 shows the response surface diagram and corresponding contour map of the compressive strength, which reflected the interactions between LCFS content and water-binder ratio under the GBFS content of 50%. Under a lower water-binder ratio, the compressive strength of mortar samples showed a slow increase with the increase in the LCFS content. As the LCFS content increased from 10% to 20%, the compressive increased by 8.9 MPa. As the LCFS content increased from 20% to 30%, the compressive strength increased by 18 MPa. However, as the water-binder ratio increased, the compressive strength of mortar samples showed a trend of rapid decline. This indicates that Ca(OH)_2_ was generated rapidly by CaO and C_2_S hydration, a large amount of OH^−^ was in the solution which provided a well alkaline environment. They promoted the Si-O, Al-O bond breaking, which was in the vitreous structure of GBFS. Finally, with the hydration process deepened, and many more hydration products were generated. This guarantees the compressive strength of the mortar samples. When the water-binder ratio increased, lots of free water was in the structure which lead to an increase in porosity and reduction in compressive strength.

Figure 8 shows the response surface diagram and corresponding contour map of the compressive strength, which reflected the interactions between GBFS content and water-binder ratio under the LCFS content of 10%. When the water-binder ratio was 0.3, with the increase of the GBFS content, the compressive strength first increased and then decreased. The contours were parabolic, which indicates that, with increasing GBFS content, more active ingredients can participate in hydration, which can increase compressive strength. However, as the hydration process continued, the quantity of hydration products decreased due to Ca(OH)_2_ content being insufficient. This cannot depolymerize the GBFS. Therefore, the compressive strength decreased. Moreover, it can be seen that GBFS content was 50%. With the increase of water-binder ratio, the compressive strength significantly decreased because the OH^−^ concentration declined and there was more water in the solution, which impeded the hydration reaction. These all has a negative impact on the compressive strength.

### 3.3. Hydtation Degree Analysis

Figure 9 shows the heat flow curve and cumulative hydration heat curve of paste samples. It can be seen that the main heat evolution peaks and the cumulative hydration heat of the LCFS-GBFS composite cementitious materials were higher than those of the cementitious materials with no LCFS. Figure 9a shows that the heat flow curves of composite cementitious materials had two main exothermic peaks which included the dissolved exothermic peak and hydration exothermic peak. The dissolved exothermic peak was the first exothermic peak, which was mainly caused by the mass dissolution of ions from the surface of the mineral immediately when the raw material reacted with water. The higher the proportion of LCFS in cementitious materials, the larger the exothermic peak value and the earlier the appearance time. This result indicates that the addition of LCFS has a certain effect on the hydration of cementitious, and was related to the large amount of active minerals in LCFS. The second exothermic peak also showed that the higher content of LCFS, the earlier hydration exothermic peak appeared, but its peak value was not bigger. This was related to the initial hydration ability of C_2_S and C_3_S, which constitutes the main mineral phase of LCFS. Moreover, the depolymerization of GBFS produced a lot of active (H_2_SiO_4_)^2−^ and (H_2_AlO_3_)^−^; they react with Ca^2+^ in the solution and SO_4_^2−^ provided by FGDG to form C-S-H gel and ettringite, which also released chemical reaction heat.

As shown in Figure 9b, the accumulated hydration heat of composite cementitious materials had a slow-growth period, which was related to the peak intensity of the exothermic peak in hydration exothermic curves. This indicates that the higher the content of LCFS, the larger quantity of Ca(OH)_2_ and C-S-H gels were generated. The uncrystallized OH^−^ and Ca^2+^ leave the solution in a highly alkaline state, which promotes the dissolving of GBFS to form C-S-H gels. These factors all cause the accumulated hydration heat to increase when the LCFS content increases.

### 3.4. XRD Analysis

Based on compressive strength performance, the improvement of various properties of the LCFS-GBFS composite cementitious materials leads to the generation of specific hydration products during the hydration process. To investigate the effect mechanism, the hydration products of the LCFS-GBFS composite cementitious materials were analyzed by using XRD. The XRD pattern of 28 days paste samples are shown in Figure 10. It can be seen that there are significant diffraction peaks of Ca(OH)_2_, as well as the diffraction peak of un-hydrated C_2_S, C_3_S. The intensity for the diffraction peak of Ca(OH)_2_, C_2_S, and C_3_S increased gradually when the LCFS content increased. This indicates that more Ca(OH)_2_ will be produced by the CaO, C_2_S, and C_3_S which were in the LCFS. It provided a good alkaline environment for the composite cementitious system, which promoted the depolymerization of GBFS. However, the diffraction peak of Ca(OH)_2_ for the sample LCFS-0 was caused by the SSM hydration; there were certain amount f-CaO and C_2_S in it. As can be observed, there existed a diffraction peak of CaSO_4_·2H_2_O, and the intensity of the diffraction peak became lower as the LCFS content increased. This indicated that the LCFS hydration can consume FGDG; the superposition effect of LCFS and FGDG can promote the decomposition of GBFS. Therefore, a large amount of C-S-H gel and ettringite were generated, which is in accord with the trend of the intensity which belonged to the diffraction peak of ettringite and C-S-H. A characteristic diffraction peak of plombierite can also be seen, which was usually attributed to C-S-H groups by alkali-activated slag. This shows that, with increased LCFS content, the whole cementitious system can produce more C-S-H gels. Moreover, the characteristic diffraction peak of C-S-H appeared between 30–32° [38]. The diffraction peak was weak and presented a bulge, which indicated that the C-S-H has low crystallization degree and different morphology, and this is also related to C_2_S, which was incomplete hydration.

### 3.5. FTIR Analysis

In order to further explore the hydration characteristics of LCFS-GBFS composite cementitious materials, especially the synergistic reaction of the raw material, FTIR was used to characterize the chemical structure of the paste samples [39]. Figure 11 shows the FTIR analysis of paste samples with different LCFS content at 28 days. It can be seen that the wave peaks near 3752, 3405, 1632, 1422, 982, 878, and 520 cm^−1^ existed in the spectrum of the samples. The absorption band of 3752 cm^−1^ was the stretching vibration band of Ca-OH, and the wave peak clearly became sharp with increased LCFS content. This indicates that the higher proportion of LCFS in cementitious materials, the more Ca(OH)_2_ were generated. This result is consistent with the XRD analysis. The absorption band of 3405 cm^−1^ was the stretching vibration band of H-O-H in water molecules, the 1635 cm^−1^ bond was the bending vibration, and these all belonged to ettringite. As shown in the spectra, it can be inferred that the wave peak became sharp and the tended to get smaller. This indicates that the free water transformed into crystalline water in the hydration process, then we generated the ettringite, the quantity, and rate of the production for ettringite, which were related to the LCFS content. The 982, 878, and 520 cm^−1^ bands were the typical absorption bands of C-S-H gels. In particular, the 982 cm^−1^ peak was the asymmetric stretching peak of Si-O-Si in the C-S-H gel, and the 878 cm^−1^ peak was the bending vibration peak of Si-OH in the C-S-H gel. The spectra shows that the peak became sharp and narrow, which indicates that the C-S-H gels increased gradually and tended to crystallize. Moreover, the 520 cm^−1^ peak was the bending vibration peak of O-Si-O in the C-S-H gel. This may be caused by the GBFS dissolved in alkaline environment which have produced C-S-H gels.

### 3.6. Solid State NMR

The structure of cement hydration products directly affect the properties of cement-based materials, and the C-S-H gels and ettringite are the most important hydration products of cement-based materials. Therefore, many scholars used NMR to study the structure and composition of the C-S-H gels and ettringite.

In cement-based minerals, the Si atom mainly existed in the Si-O tetrahedral pattern, and the Q^n^ represented the polymerization state of Si-O tetrahedron [40]. Moreover, the chemical shift of ^29^Si varied with the degree of polymerization of C-S-H gels which were in the solid silicate. The experimental result at 7 days of the ^29^Si MAS NMR spectra is shown Figure 12a, it can be found from the spectra that with the ^29^Si MAS NMR spectral peak position shifted slightly to the right, which indicated that the silicon oxygen network structure in the cementitious structure changed correspondingly. Much more highly polymerized C-S-H gels were generated as the LCFS content increased. The peaks at approximately −73.6 ppm and −74.9 ppm were assigned to Q^0^, it can be seen that the peak became significant as the LCFS content increased. This indicates that there existed un-hydrated C_2_S and C_3_S, which were in accord with XRD and FTIR results. The peaks at approximately −78.6 ppm and -80.2 ppm were assigned to Q^1^, and the Q^1^ was the typical formant of C-S-H gel, which because of the existence pattern of Si-O tetrahedron in C-S-H gel were mainly Q^1^ and Q^2^ structure. Through the peak area of the samples by chemical shift at −78.6 ppm and −80.2 ppm, the hydration degree can be analyzed qualitatively even without deconvolution. This also indicated that a large amount of C-S-H gel was generated as the LCFS content increased. The peaks at approximately −84.3 ppm was assigned to Q^2^, and multiple peaks existed nearby. This means that LCFS promoted the Si-O bond to depolymerize those that were in the GBFS, and then the Si in the system gradually developed to a higher polymerization state.

Figure 12b shows the experimental result of the ^27^Al MAS NMR spectra at 7 days. From the spectra, the chemical shift of ^27^Al 4 coordination was 50 to 74 ppm, which was regarded as C-S-H gel. The chemical shift of ^27^Al 6 coordination was −8 to 12 ppm, which was regarded as ettrigite. As shown in the spectra, the absorption peaks appeared at chemical shift value of C-S-H gel for all samples. However, the quantity of the peaks increased when the LCFS content increased. These indicates that the C-S-H gel was generated in all cementitious, but much more C-S-H gels could be generated by the hydrated C_2_S in LCFS and SSM. The relative intensity of the absorption peaks for ^27^Al 6 coordination increased when the LCFS content increased, but this is not obvious. This indicates that the ettrigite was generated mainly by GBFS hydration. A large amount of (H_3_AlO_4_)^2−^ was dissolved from GBFS, then reacted with SO_4_^2−^ which was in the gypsum. Above all, LCFS was the effective activator for GBFS which can promote the dissolving of GBFS.

### 3.7. SEM Analysis

The SEM images of paste samples with different LCFS content at 28 days are shown in Figure 13. As can be observed from Figure 13a, the micrograph of paste sample shows a large amount of flocculent C-S-H gels and bar-shaped ettringite. However, the structure of the cementitous system was relatively loose with a large number of micro-pores. As the LCFS content increased, more C-S-H gels appeared in the Figure 13b. They encased the ettringite and made the micro-pores decrease. This also made the structure of the cementitous system look tighter. Then a denser structure of the cementitous system is shown in Figure 13c. It was clear that the ettringite were almost covered with C-S-H gels. Finally, almost no micro-pores can be observed in Figure 13d, where the C-S-H gels and ettringite intertwined. Compared to the sample LCFS-0, the amount, size, and dense structure in LCFS-GBFS paste samples were significantly better. This indicates that much more hydration products can be produced as the LCFS content increased. This is consistent with the above analytical results.

## 4. Applications

The characteristics of LCFS-GBFS composite cementitious materials indicates that it has some application potential in construction fields. The results of this study show some similarities with cement-based materials. First, the mechanical properties of LCFS-GBFS composite cementitious materials are similar to those of cement-based materials, which shows potential for its application in the construction field. Secondly, the main hydration products are C-S-H gels and ettringite, and these are also similar to the hydration products of cement-based materials, which provides a theoretical basis for its application.

Overall, the applications of LCFS-GBFS composite cementitious materials have many potential development directions. First of all, it can be used to produce road concrete and in road construction in industrial parks. Because LCFS plants are mostly located in the areas with poor transportation conditions. Secondly, it can be used to produce prefabricated concrete parts, lightweight wall material, and bricks. Finally, it can be used as backfill material for mining. In addition, there are some contributions with reducing CO_2_ emissions and construction costs because its application in construction engineering can reduce the use of cement.

## 5. Conclusions

This paper studied the possibility of using LCFS as an activator to prepare composite cementitious materials. Through Box-Behnken design of response surface and strength test, the effect of the content of LCFS on compressive strength was evaluated. Moreover, using a variety of microstructural testing methods, the hydration products and micro-morphology of LCFS-GBFS composite cementitious materials were understood preliminarily. The main conclusions from the experimental results are as follows:(1)The LCFS has a hydration superposition effect to the composite cementitious materials, the compressive strength of the mortar samples by using LCFFS are significantly higher than the sample without LCFS.(2)The compressive strength of the mortar samples at 3, 7, and 28 days are 26.6, 35.3, and 42.7 MPa, respectively, when the LCFS-GBFS-SSM-FGDG ratio is 3:5:1:1 and the water-binder ratio is 0.3. The LCFS-GBFS composite cementitious materials have similar mechanical properties to cement-based materials.(3)Through the response surface design experiment, the ratio of LCFS-GBFS is the most important factor which can affect the compressive strength. The second factor is water-binder ratio.(4)The hydration products of LCFS-GBFS composite cementitious materials are C-S-H gels, and ettringite. The preliminary hydration mechanism of LCFS-GBFS composite cementitious materials are as follows: Firstly, the hydration of C_2_S, C_3_S in LCFS generates Ca(OH)_2_, C-S-H gels and many Ca^2+^ and OH^−^, they increase the alkalinity of the solution. Subsequently, the GBFS dissolve and release a large amount of (H_2_SiO_4_)^2−^, (H_2_AlO_3_)^−^, and react with Ca^2+^ in the solution and SO_4_^2−^, which is dissolved by FGDG, then a large quantity of C-S-H gels and ettringite are generated. As the hydration process deepens, the minerals of SSM gradually dissolve, which promotes the further hydration of GBFS, and generates the hydration products. Finally, the synergistic hydration of LCFS-GBFS-SSM-FGDG is continuous, which promotes the production of more hydration products to fill the pore structure of cementitious system. This results in good compressive strength performance of the LCFS-GBFS composite cementitious materials.

In general, the LCFS-GBFS composite cementitious materials were prepared without cement, and it is better for reducing CO_2_ emissions than cement-based materials. Meanwhile, it also has good mechanical performance. Based on the preliminary study of the LCFS-GBFS composite cementitious materials, the potential feasibility of LCFS-GBFS composite cementitious materials application in construction engineering is proved. However, the hydration mechanism and mechanical properties of LCFS-GBFS composite cementitious materials still need to be further studied, including the single mineral test in LCFS and the synergistic reaction of cementitious materials. Moreover, in order to verify its durability, more construction projects should be implemented by using LCFS-GBFS composite cementitious materials, such as road, bricks, and prefabricated parts, etc.

## Figures and Tables

**Figure 1 materials-16-02385-f001:**
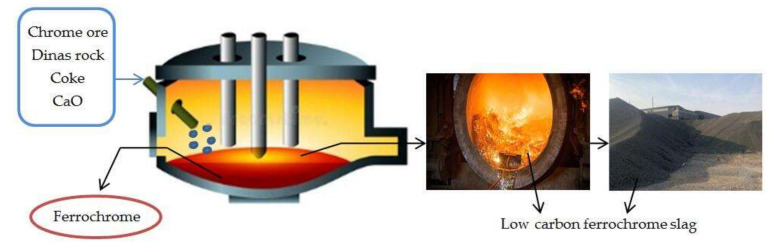
The production process of low carbon ferrochrome slag.

**Figure 2 materials-16-02385-f002:**
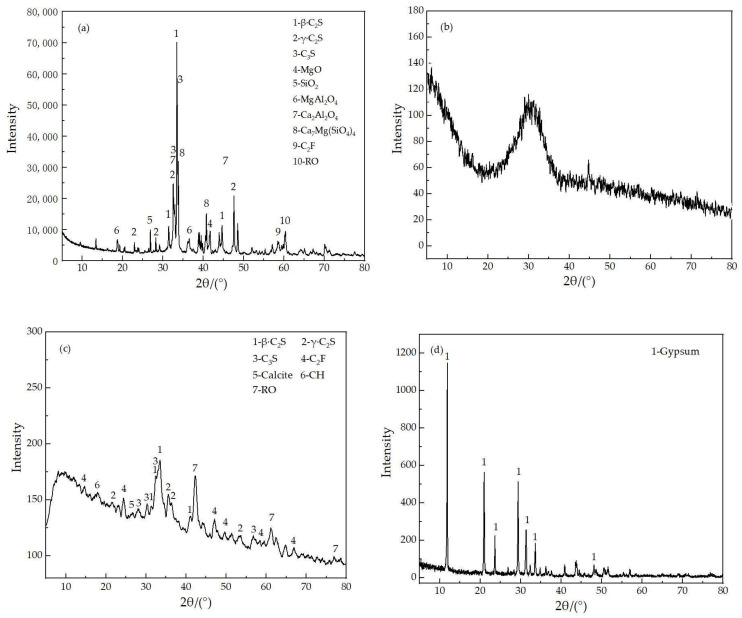
XRD patterns of raw materials: (**a**) LCFS; (**b**) GBFS; (**c**) SSM; (**d**) FGDG.

**Figure 3 materials-16-02385-f003:**
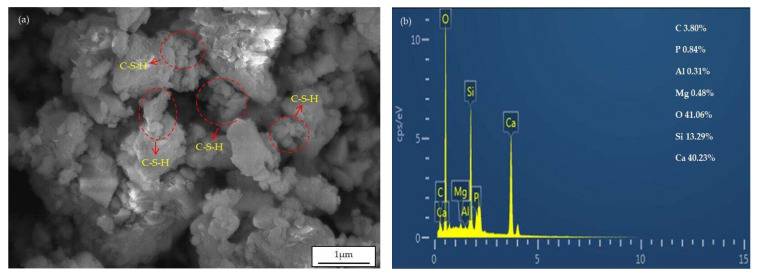
The micrograph and EDS spectra of LCFS: (**a**) The micrograph of LCFS; (**b**) The EDS of C_2_S on (**a**).

**Figure 4 materials-16-02385-f004:**
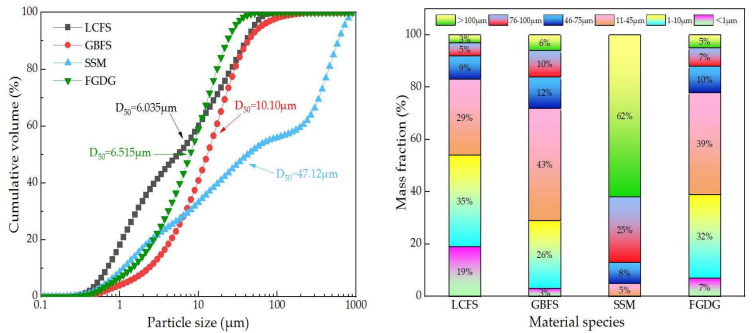
Particle size distribution of raw materials and the percentage of particle size.

**Figure 5 materials-16-02385-f005:**
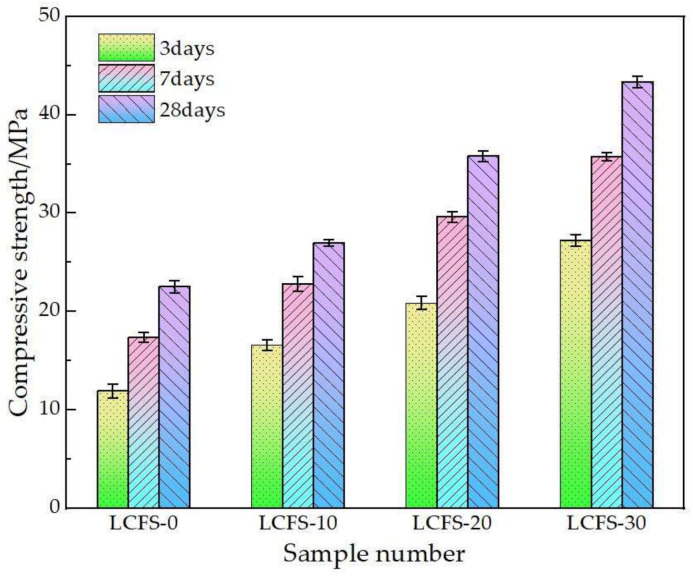
Compressive strength of mortar samples with different LCFS-GBFS content.

**Figure 6 materials-16-02385-f006:**
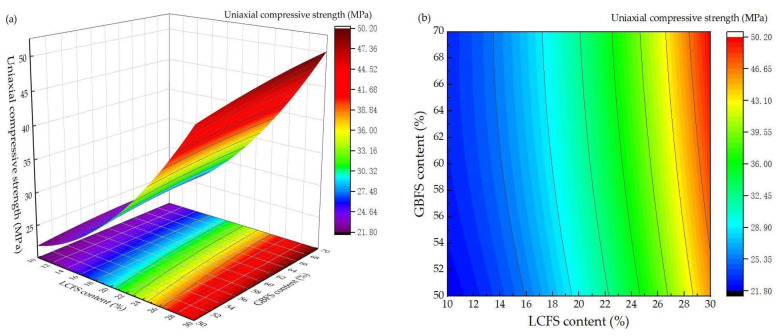
Effect of interaction between LCFS and GBFS content on compressive strength: (**a**) response surface diagram; (**b**) contour map.

**Figure 7 materials-16-02385-f007:**
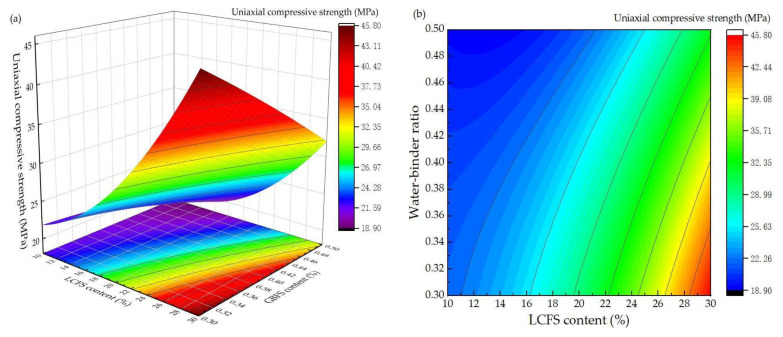
Effect of interaction between LCFS content and water-binder ratio on compressive strength: (**a**) response surface diagram; (**b**) contour map.

**Figure 8 materials-16-02385-f008:**
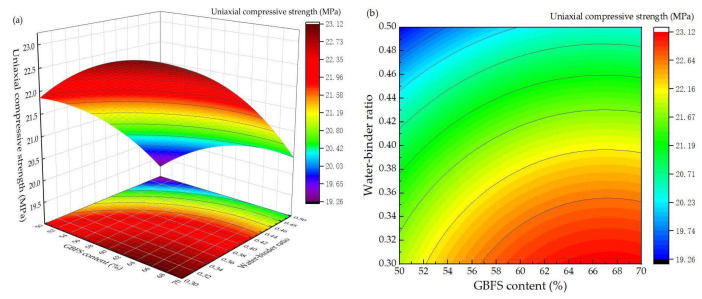
Effect of interaction between GBFS content and water-binder ratio on compressive strength: (**a**) response surface diagram; (**b**) contour map.

**Figure 9 materials-16-02385-f009:**
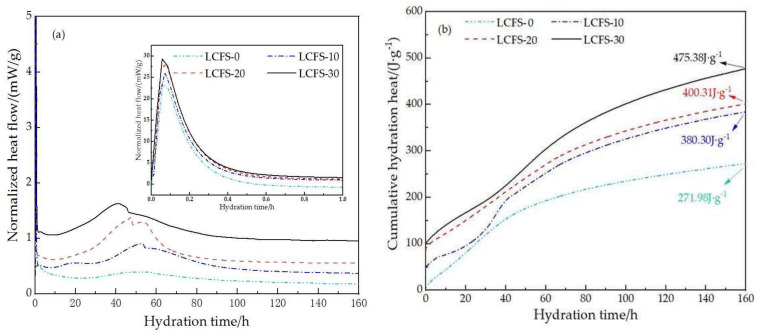
The hydration heat of paste samples: (**a**) heat flow curves; (**b**) cumulative hydration heat.

**Figure 10 materials-16-02385-f010:**
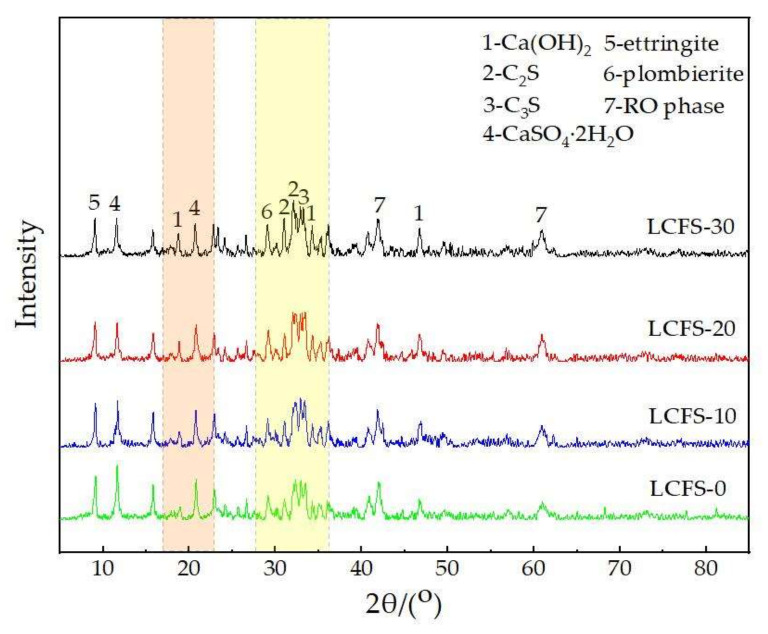
XRD patterns of LCFS-GBFS composite cementitious materials pastes at 28 days.

**Figure 11 materials-16-02385-f011:**
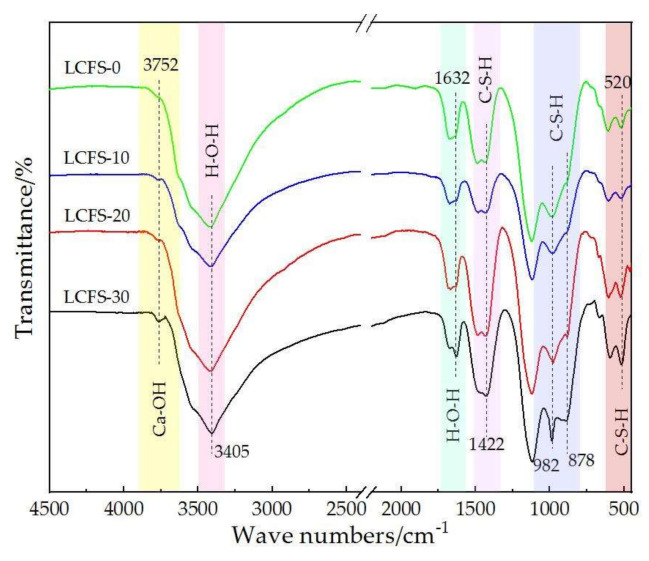
FTIR analysis of the samples with different LCFS content at 28 days.

**Figure 12 materials-16-02385-f012:**
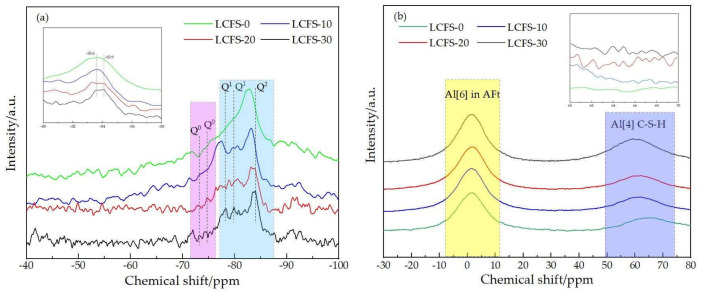
NMR spectra of the paste samples at 28 days: (**a**) 29Si MAS-NMR; (**b**) 27Al MAS-NMR.

**Figure 13 materials-16-02385-f013:**
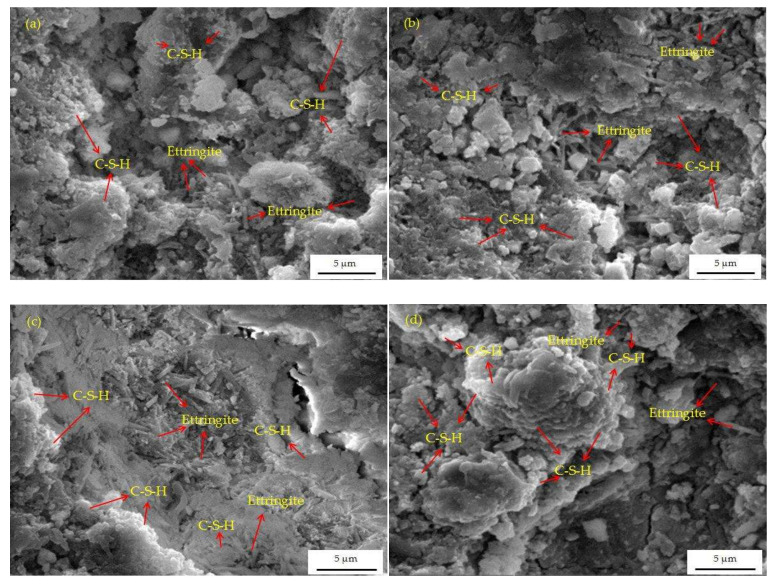
The SEM micrograph of paste samples with different LCFS content at 28 days: (**a**) LCFS-0; (**b**) LCFS-10; (**c**) LCFS-20; (**d**) LCFS-30.

**Table 1 materials-16-02385-t001:** Chemical composition of LCFS, GBFS, SSM and FGDG (wt%).

Composition	LCFS	GBFS	SSM	FGDG
CaO	50.53	38.69	33.65	32.06
SiO_2_	33.21	36.35	12.36	1.98
Al_2_O_3_	9.53	16.23	1.97	0.77
MgO	1.02	9.27	5.31	1.16
Na_2_O	-	0.52	0.26	0.08
Fe_2_O_3_	1.07	1.05	33.42	0.39
MnO	0.16	0.53	5.06	0.03
TiO_2_	0.60	0.79	0.89	0.06
K_2_O	0.08	0.36	0.22	0.13
SO_3_	0.08	-	0.13	43.89
P_2_O_5_	-	-	1.62	-

**Table 2 materials-16-02385-t002:** The mix proportions of LCFS-GBFS cementitious materials.

Mix Code	LCFS	GBFS	SSM	FGDG
LCFS-0	0	80	10	10
LCFS-10	10	70	10	10
LCFS-20	20	60	10	10
LCFS-30	30	50	10	10

**Table 3 materials-16-02385-t003:** Code and level of response surface design factors.

Level	X1	X2	X3
−1	10	50	0.3
0	20	60	0.4
1	30	70	0.5

## Data Availability

The data that support the findings of this study are available from the corresponding author on reasonable request.

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
