# Peer review of "Preparation and Hydration Mechanisms of Low Carbon Ferrochrome Slag-Granulated Blast Furnace Slag Composite Cementitious Materials"

_materials, 2023, doi:10.3390/ma16062385_

Round 1
Reviewer 1 Report
Abstract:
Need to add a conclusions and suggestion for future recommendations at the last sentence in abstract.
Method
Please state the quantity of low carbon ferrochrome slag, granulated blast furnace slag, steel slag mud and flue gas desulfurization gypsum used in the experiment.
2.2.3: Please add information for X-ray Diffraction (XRD) used such as type of model and country production, likely t (model Bruker-AXS D8 Advanced, USA).
Is there control specimen (mixture of cement and sand) that is being used to compare the mechanical and physical properties with LCFS, GBFS, SSM and FGDG?
Result
Please discuss current findings using LCFS, GBFS, SSM and FGDG as cementitious materials with previous studies.
Based on results, discuss the application of using LCFS, GBFS, SSM and FGDG in construction purposes.
Conclusion
State recommendation for future study at the end of the conclusion.
Author Response
Dear reviwer:
Thank you for your decision and constructive comments on my manuscript. We have carefully considered the suggestion of Reviewer and make some changes. We have tried our best to improve and made some changes in the manuscript.
The red part that has been revised according to your comments. Revision notes, point-to-point, are given as follows:
Point 1: Need to add a conclusions and suggestion for future recommendations at the last sentence in abstract.
Response 1: According to the suggestion, we have added a conclusion and a suggestion for future recommendations at the last sentence in abstract. The specific content is described as “ This paper can provide a preliminary theoretical foundation for the development of LCFS-GBFS composite cementitious materials. Moreover, the deep hydration mechanism analysis and engineering application should be studied in the future.”
Point 2: Please state the quantity of low carbon ferrochrome slag, granulated blast furnace slag, steel slag mud and flue gas desulfurization gypsum used in the experiment.
Response 2: Dear reviwer, We need to explain to you why there is no quantity of cementitious material ( low carbon ferrochrome slag, granulated blast furnace slag, steel slag mud and flue gas desulfurization gypsum ), this is because the mass percentage of the cementitious material is specified in the mix proportion. And the total mass of the cementitious material is determined to be 450g. Therefore, the quatity of each cementitious material is not continued to be listed, so as to avoid repeating the content with Table 2. I hope to get your understanding and recognition.
Point 3: 2.2.3: Please add information for X-ray Diffraction (XRD) used such as type of model and country production, likely t (model Bruker-AXS D8 Advanced, USA).
Response 3: We have added information for X-ray Diffraction in 2.2.3.
Point 4: Is there control specimen (mixture of cement and sand) that is being used to compare the mechanical and physical properties with LCFS, GBFS, SSM and FGDG?
Response 4: We are very sorry to tell you that there is no control specomen (mixture of cement and sand) experiment due to the following two factors: first, the initial purpose of our experiment is only to verify whether the LCFS-GBFS composite cementitious materials has good mechanical properties, meanwhile, the hydration mechanism analysis part only tests the LCFS-GBFS composite cementitious materials. Secondly, considering the mechanical and physical properties of cement mortar are well known, so we did not carry out experiment on it.
Point 5: Please discuss current findings using LCFS, GBFS, SSM and FGDG as cementitious materials with previous studies.
Response 5: We have made a preliminary discussion for the findings by using LCFS, GBFS, SSM and FGDG as cementitious materials,which are after each test result in manuscript. However, we will explain to you again that there is no previous studies about using LCFS, GBFS, SSM and FGDG as composite cementitious materials. The LCFS are mostly disposed in landfill, and the others are used as a cement mixture. We also accidentally found that these has good gelling properties. So we did not have some previous studies to compare and discuss, and there are a lot of further researchs to be done.
Point 6: Based on results, discuss the application of using LCFS, GBFS, SSM and FGDG in construction purposes
Response 6: As your kind suggestion, we have added the section 4 to discuss the application of using LCFS-GBFS composite cementitious materials in construction.
Point 7: State recommendation for future study at the end of the conclusion.
Response 7: According to your suggestion, we have added the recommendation for future study at the end of the conclusion.

Reviewer 2 Report
The article presents a study on Hydration Mechanism of Low Carbon Ferrochrome Slag-Granulated Blast Furnace Slag Composite Cementitious Materials. Although it has interesting information requires some corrections:
- Include in the abstract the main conclusions obtained in your research based on the results obtained.
- Introduction: “Ferrochrome Slag is a Gray-Black Metallurgical Slag Which is Produced During Smelting Caobon Ferrochrome Alloy With Different Carbon Content. And the metallurgical WASTE SLAG is FORMED after High Temperatuure Reducion of Chromite by Pyroelectric Furnace at 1700 ℃ Which Use Carbon as Reducing Agent [1]. According to the carbon content, it can be divided into low carbon and high carbon ferrochrome slag. The High Carbon Ferrochrome Slag is Generally Agglomerate and the Low Carbon Ferrochrome Slag is Usually Small Particles. In 2022, The Carbon Ferrochrome Slags Were Produced 14.62 thousand ”Please, after this section includes works published with Ferrochrome Slag as construction materials, replacing OPC or in an alkaline activation process. Note that your work is not the first to propose an application of Ferrochrome Slag. So please make clear to readers of the article what has already been researched with this material and what still needs to be discovered. What are the knowledge gaps studied in your research?
- “The Granulated Blast Furnace Slag is the Industrial Waste Which is Discharged in the Process of Iron Smelting. AS A Typical Pozzolanic Material, granulated Blast Furnace Slag is Formad After Water Quenching and Extremély Cold Treatment, It has Vitreous Structure and Potential Activity Which is Widiel Used Increte and Concrete Industry. ” I do not agree that this material is considered pozolanic. Observe the classic definition of pozolanical materials: materials predominantly rich in aluminosilicate. This is not the case of GBFS, which generally presents mostly calcium. It is a cementitious material, a lesser reactivity, but not a pozzolanic material. See articles about GBFs and discuss this information, I suggest analyzing articles: 10.1016/J.CSCM.2021.E00723 and 10.3390/MA14082069.
- My main criticism of the work is: There is research that has already evaluated composite cementitious materials are prepared in this Word, Which Use low carbon ferrochrome slag and granulated Blast Furnace Slag (GBFS) as Main Raw Material, Using Steel Slag Mud (SSM) and Flow Gas Desulfiliation Gypsum (FGDG) as the activator or the like. What are the innovations and ineditism of the research? Please make it clear in your manuscript.
-“and Resulted in the formation of more rapid Products, Special Ettringite and C-S-H Products.” Correct by removing the repetition of the word products.
-“It indicates that the ca (OH) 2 was Generated Qiuckly by LCFS Hydration Which Promoted the Decomposition of GBFS, and Resulted in the formation of more rapid Products, Special Etthingeritis and C-S-H Products” What are the evidence that happened? How can LCFS promote this decomposition? Explain.
-"Moreover, C-S-H Gels Were Also Mass Generated by C2S in LCFS, Which Filled the Pores and Enhanced The Early Compressive Strength of Morte Samples." It doesn't seem right to me. C2S is known to be responsible for resistance at more advanced ages. How can this be explained?
- Figure 4: The authors do not provide a satisfactory explanation for the effect of LCFs on compression resistance. At the end of the discussion of this result should it be clear to the reader: Why does LCFS promote material resistance? Compare the results with similar searches.
- Improve the quality of figures 3 and 8.
- Figure 8 (a): between 40 and 60 h there is a variation in the heat flow of the compositions. What justifies this behavior? Usually after the initial peak the heat flow is stable. Because behavior is different for the compositions used in this research.
- Please review your conclusions based on previous comments.
Author Response
Dear reviewer:
Thank you for your comments and suggestion concerning our manuscript. The comments and suggestions are all professional and very helpful for revising and improving our paper, as well as the important guiding significance to our researches. We have studystudied comments carefully and have made correction which we hope meet with approval. Revision notes, point-to-point, are given as follows:
Point 1: Include in the abstract the main conclusions obtained in your research based on the results obtained.
Response 1: As your kind suggestion, we have added the main conclusions at the end of the abstract.
Point 2: Introduction: “Ferrochrome Slag is a Gray-Black Metallurgical Slag Which is Produced During Smelting Caobon Ferrochrome Alloy With Different Carbon Content. And the metallurgical WASTE SLAG is FORMED after High Temperatuure Reducion of Chromite by Pyroelectric Furnace at 1700 ℃ Which Use Carbon as Reducing Agent [1]. According to the carbon content, it can be divided into low carbon and high carbon ferrochrome slag. The High Carbon Ferrochrome Slag is Generally Agglomerate and the Low Carbon Ferrochrome Slag is Usually Small Particles. In 2022, The Carbon Ferrochrome Slags Were Produced 14.62 thousand ”Please, after this section includes works published with Ferrochrome Slag as construction materials, replacing OPC or in an alkaline activation process. Note that your work is not the first to propose an application of Ferrochrome Slag. So please make clear to readers of the article what has already been researched with this material and what still needs to be discovered. What are the knowledge gaps studied in your research?
Response 2: According to your request, we have added the corresponding content in the last paragraph of this section.
Point 3: “The Granulated Blast Furnace Slag is the Industrial Waste Which is Discharged in the Process of Iron Smelting. AS A Typical Pozzolanic Material, granulated Blast Furnace Slag is Formad After Water Quenching and Extremély Cold Treatment, It has Vitreous Structure and Potential Activity Which is Widiel Used Increte and Concrete Industry. ” I do not agree that this material is considered pozolanic. Observe the classic definition of pozolanical materials: materials predominantly rich in aluminosilicate. This is not the case of GBFS, which generally presents mostly calcium. It is a cementitious material, a lesser reactivity, but not a pozzolanic material. See articles about GBFs and discuss this information, I suggest analyzing articles: 10.1016/J.CSCM.2021.E00723 and 10.3390/MA14082069.
Response 3: We qiute agree with you on the definition of GBFS and studied the article you provided. We have corrected the statement in the text. I need to explain to you that maybe there is something wrong with the way we are presenting it, and our intention is that the GBFS is a typical material with pozzolanic activity, but not pozzolanic material.
Point 4: My main criticism of the work is: There is research that has already evaluated composite cementitious materials are prepared in this Word, Which Use low carbon ferrochrome slag and granulated Blast Furnace Slag (GBFS) as Main Raw Material, Using Steel Slag Mud (SSM) and Flow Gas Desulfiliation Gypsum (FGDG) as the activator or the like. What are the innovations and ineditism of the research? Please make it clear in your manuscript.
Response 4: We also appreciated your kind criticism and suggestion. We have added innovation points and some deficiencies, and they are at the end of the introduction and the conclusions of the manuscript. Moreover, I will also introduce the innovations and deficiencies of this study to you here. As you konw, LCFS are commonly added to cement as an admixture, and the addition is limited significantly. Therefore, the most obvious innovation in this study is the preparation of composite cementitious materials by using LCFS, GBFS, SSM and FGDG as raw materials, and without cement. The application of this composite cementitous materials can reduce the amount of cement and contribute to the reduction of CO2 emissions significantly. Meanwhile, it is worth noting that there are some deficiencies of this study. This paper is only a preliminary study on LCFS-GBFS composite cementitious materials, and because of the mineral composition of the raw materials is more complex than cement, then the hydration mechanism of LCFS-GBFS composite cementitious materials still need to be further studied, including the single mineral test in LCFS and the synergistic reaction of cementitious materials.
Point 5: “and Resulted in the formation of more rapid Products, Special Ettringite and C-S-H Products.” Correct by removing the repetition of the word products.
Response 5: According to your suggestion, we have removed the repetitive word products.
Point 6: “It indicates that the Ca(OH)2 was Generated Qiuckly by LCFS Hydration Which Promoted the Decomposition of GBFS, and Resulted in the formation of more rapid Products, Special Etthingeritis and C-S-H gels” What are the evidence that happened? How can LCFS promote this decomposition? Explain.
Response 6: According to your request, I need to explain to you that why does this reaction exist. The mineral composition in LCFS mainly includes C3S,C2S and f-CaO, they hydrate to produce Ca(OH)2, but the liquid phase still contains many Ca2+ and OH-, then let the solution in a high alkalinity state. The vitreous in GBFS is mainly silicon oxygen tetrahedron structure, and the degree of polymerization is low. The alkalinity of liquid phase system can promote the depolymerization of silico-oxygen tetrahedron and al-oxygen tetrahedron in GBFS, then produce a lot of active (H2SiO4)2- and (H2AlO3)-, they react with Ca2+ in the solution and SO42- provided by FGDG to form C-S-H gel and ettringite.
Point 7: "Moreover, C-S-H Gels Were Also Mass Generated by C2S in LCFS, Which Filled the Pores and Enhanced The Early Compressive Strength of Morte Samples." It doesn't seem right to me. C2S is known to be responsible for resistance at more advanced ages. How can this be explained?
Response 7: We have corrected the typescript error C2S, and the C3S is correct. However, we also think that C2S may contribute to early hydration in this system, because of the specific surface area of LCFS is 915 m2/kg, which is nearly 2.5 times to cement. This may mean that crystal lattice of the minerals in the LCFS is distorted, which can affect the rate of hydration reaction for minerals. In addition, some very fine CaO particals are also attached to the C2S and C3S, they can also speed up the rate of hydration, when the concentration of Ca2+ and OH- in the liquid phase does not reach the solubility product of Ca(OH)2 crystal, the Ca2+ react with (H2SiO4)2- to form C-S-H gels firstly. This is also the focus of our future study.
Point 8: Figure 4: The authors do not provide a satisfactory explanation for the effect of LCFs on compression resistance. At the end of the discussion of this result should it be clear to the reader: Why does LCFS promote material resistance? Compare the results with similar searches.
Response 8: According to your request, we have revised some statements in the text, and added the theoretical basis to demonstrate the positive effect of LCFS on compressive strength.
Point 9: Improve the quality of figures 3 and 8.
Response 9: We have improved the quality of Figure 3 and Figure 8.
Point 10: Figure 8 (a): between 40 and 60 h there is a variation in the heat flow of the compositions. What justifies this behavior? Usually after the initial peak the heat flow is stable. Because behavior is different for the compositions used in this research.
Response 10: According to your comment, We would like to share with you some preliminary views on the characteristics of heat flow curves of composite cementitious materials. As we all know, the hydration of the cementitious materials generates hydration products and releases chemical reaction heat, the exothermic process is very stable on the heat flow curve. However, with the increase of LCFS content, the second exothermic peak of composite cementitious materials fluctuated obviously. This may include several factors as follows: first, the mineral composition of composite cementitious materials is more complex than that of cement-based material, these lead to the possible formation of a variety of silicaluminate minerals in the solution, but not only C-S-H gel and ettringite, and their heat release is erratic. Secondly, the specific surface area of composite cementitious material is much smaller than that of cement, the specific surface area of LCFS is nearly 2.5 times that of cementand, then some minerals that lattice is distorted during the grinding process. This also causes some mineral to hydrate and release a lot of heat suddenly between 40 and 60 h, maybe f-CaO. Finally, it is possible that the mineral responsible for the fluctuating of heat flow curve is mainly existed in the LCFS. Therefore, the higher the LCFS content, the more obvious this performance, between 40 and 60 h. Above all, we still have a lot of further research to do on the mechanism of hydration, such as the single mineral test in LCFS.
Point 11: Please review your conclusions based on previous comments.
Response 11: According to your suggestion, we have corrected some errors in the conclusions.

Round 2
Reviewer 1 Report
all the corrections has been made. this paper can be published.
Reviewer 2 Report
The authors responded to all previous review comments.